# Chronic Hepatitis B Viral Activity Enough to Take Antiviral Drug Could Predict the Survival Rate in Malignant Lymphoma

**DOI:** 10.3390/v14091943

**Published:** 2022-08-31

**Authors:** Kwang-Il Seo, Jae-Cheol Jo, Da-Jung Kim, Jee-Yeong Jeong, Sangjin Lee, Ho-Sup Lee

**Affiliations:** 1Department of Internal Medicine, Kosin University College of Medicine, Busan 49267, Korea; 2Chang Kee-Ryo Memorial Liver Institute, Kosin University College of Medicine, Busan 49267, Korea; 3Department of Hematology and Oncology, Ulsan University Hospital, University of Ulsan College of Medicine, Ulsan 44033, Korea; 4Department of Biochemistry, Kosin University College of Medicine, Busan 49267, Korea; 5Institute for Medical Science, Kosin University College of Medicine, Busan 49267, Korea; 6Department of Statistics, Pusan National University, Busan 46241, Korea

**Keywords:** antiviral agents, extrahepatic malignancy, hepatitis B virus, liver cirrhosis, HBV DNA integration, malignant lymphoma, viral activity

## Abstract

Hepatitis B virus (HBV) infection carries a risk of liver cancer and extrahepatic malignancy. However, the incidence trend and clinical course of malignant lymphoma (ML) in HBV patients are not well known. Data about ML newly diagnosed in chronic hepatitis B (CHB) patients from 2003 to 2016 were collected from National Health Insurance Service claims. A total of 13,942 CHB patients were newly diagnosed with ML from 2003 to 2016. The number of patients increased 3.8 times, from 442 in 2003 to 1711 in 2016. The 2-year survival rate of all patients was 76.8%, and the 5-year survival rate was 69.8%. The survival rate of patients taking antivirals due to high viral activity before their diagnosis with ML was significantly lower than that of patients with lower viral activity without antivirals (1 yr—77.3%, 3 yr—64.5%, and 5 yr—58.3% vs. 1 yr—84.0%, 3 yr—73.4%, and 5 yr—68.0%, respectively). The survival rate of patients with liver cirrhosis (LC) at baseline was significantly lower than that of those without LC. Cirrhotic patients taking antivirals before ML diagnosis had a worse prognosis than who did not. High viral activity in CHB patients with ML seems to be useful in predicting the prognosis for survival.

## 1. Introduction

About 350 million people worldwide are estimated to be infected by the hepatitis B virus (HBV), and 600,000 deaths annually can be attributed to chronic hepatitis B (CHB) [1]. Viral hepatitis-related mortality now exceeds the mortality attributed to HIV, tuberculosis, and malaria [2]. HBV is a major cause of liver cirrhosis (LC) and hepatocellular carcinoma (HCC) [3]. Since the introduction of potent antiviral drugs, CHB patients have been able to achieve effective viral suppression and survive longer. As a result, various comorbidities have increased, including diabetes mellitus (DM), hypertension (HTN), and chronic kidney disease (CKD) [4]. In addition, HBV infection carries a risk of extrahepatic malignancy [5]. A typical extrahepatic malignancy associated with HBV is malignant lymphoma (ML) [6]. Because the prevalence of HBV is high in Asia, the number of ML patients with HBV is likely to increase in the future [7].

Currently, CHB patients are prescribed antivirals according to guidelines based on HBV-DNA elevation and underlying liver function [8]. In other words, antiviral treatment depends on viral activity and related liver status. After introduction of entecavir (ETV) and tenofovir (TDF), the risks of extrahepatic malignancy seem to be increasing as a result of the prolonged survivals [5]. However, clinical data about the incidence trend and natural course of ML in HBV since introduction of potent antivirals are limited. This study was performed to determine how HBV infection and antiviral treatment affect the development and prognosis of ML. In addition, we investigated whether ML development could be affected by the kinds of antivirals.

## 2. Materials and Methods

### 2.1. Study Design (Population)

Korea has a single payer, universal health insurance system that covers more than 99% of its population. The National Health Insurance Service (NHIS) has a comprehensive health database of both inpatients and outpatients. It includes all medical claims data for prescription drugs, codes of diagnosis and treatment, and all reimbursed medical procedures. In addition, the claims data include information such as age, sex, accompanying diseases, and medications. This study on HBV was conducted using insurance claims from the NHIS containing any codes referencing the diagnosis or treatment of CHB and related disease in Korea, an HBV-endemic country. The Institutional Review Board/Ethics Committee of “Blinded for peer review” approved this study and granted a waiver of informed consent because of the retrospective study design (“Blinded for peer review”). This study was performed in accordance with local law and the Declaration of Helsinki. All authors had access to the study data and reviewed and approved the final manuscript.

We collected data for all CHB patients in Korea who were newly diagnosed with ML between 1 January 2002 and 31 December 2016. As a control group, we used the propensity score matching method to collect data on 10 times that number of CHB patients who did not develop ML (Appendix A). The use of antivirals was determined based on claim records for lamivudine, adefovir, telbivudine, ETV, and TDF. The treatment periods were calculated based on medication claim date records. Analyzing those data also allowed us to infer the severity of the accompanying disease. In that way, baseline liver function was divided into chronic hepatitis and cirrhosis. In cirrhotic patients, compensation or decompensation was determined by a severity assessment. The index date was the time when ML was diagnosed, when antiviral drugs administered, when treatment drugs administered according to liver function, when a procedure performed to treat complications, or when the patient died.

To compare the effects of antivirals (ETV vs. TDF) on ML development, we performed a subgroup analysis. In Korea, patients with CHB were given ETV from 2007 and TDF from 2012, following reimbursement policies of the NHIS. Treatment-naïve CHB patients prescribed TDF or ETV as initial treatment from 2012 to 2014 were followed up until 2017 to observe the occurrence of ML. To apply the same observation period, the follow-up durations of patients diagnosed with ML (the time taken to diagnose ML after drug administration) were randomly assigned to control patients. CHB patients who started antivirals within the first 3 months before their diagnosis of ML as prophylactic therapy were excluded.

Patients who met one or more of the following criteria were excluded: chronic hepatitis C, chronic hepatitis D, acute viral hepatitis, HIV infection, liver transplantation, malignant neoplasm in the liver including HCC and cholangiocarcinoma, and age less than 20 or more than 80 (limitation of age was used only in survival analysis) (Appendix A).

### 2.2. Data Collection

All patients were classified using the International Classification of Disease, version 10 (ICD-10) codes. CHB was defined as B18.0 or B18.1. ML was classified by subtype using ICD-10 codes. Mature B cell type was determined as C82, C83, C85, and C88.4. The mature T-cell and NK type was classified as C84, Hodgkin lymphoma as C81, and unknown as C88.0, C88.2, C88.3, C88.7, and C88.9. LC was defined as K74.0, K74.1, K74.2, and K74.6. Decompensation was defined as LC taking diuretics for ascites control, vasoconstrictors for variceal bleeding, or a non-selective beta blocker for varix. In addition, in the presence of an abdominal paracentesis code or endoscopic esophageal and gastric varices treatment code, the patient was diagnosed with decompensated cirrhosis (Appendix A). In addition to liver disease, co-morbidities used in the analyses were diagnosed as the ICD-10 codes for DM, HTN, and CKD (Appendix A). Antiviral treatment was defined as prescription codes for lamivudine, adefovir, telbivudine, ETV, or TDF for more than 6 months (Appendix A).

### 2.3. Statistical Analysis

To make the control group, we used a propensity score-matching analysis to reduce the effect of selection bias and potential confounding factors. Propensity scores were conducted using the following variables: age, sex, LC, DM, HTN, and CKD. A logistic regression model was used to compare the baseline characteristics between the ML and non-ML groups. The study endpoints were survival and the development of ML in HBV patients. The survival rates and cumulative incidence of ML were estimated using the Kaplan–Meier method and compared using a log-rank test. Univariate and multivariable Cox proportional hazards models were used to compare clinical outcomes. All statistical analyses were performed using R statistical software, version 3.5.1 (R Foundation Inc.; http://cran.r-project.org/). The propensity score-matching analysis used the “Matchit” package. Survival estimation and comparisons between groups used the “survival” package, and the cumulative incidence estimates used the “cmprsk” package. All reported *p* values are two-tailed, and *p* values less than 0.05 were considered statistically significant.

## 3. Results

### 3.1. Chronic Hepatitis B as a Risk Factor for Malignant Lymphoma

#### 3.1.1. Crude Incidence of Malignant Lymphoma in HBV Patients in Korea (2003–2016)

We analyzed data for CHB patients newly diagnosed with ML from January 2003 to December 2016 (Table 1). The number of HBV patients newly diagnosed with ML increased 3.8 times, from 442 in 2003 to 1711 in 2016 (Table 2). In the same period, the number of cumulative CHB patients doubled, from 320,182 to 646,273 (Appendix A). Over time, both the number of HBV patients newly diagnosed with ML and the proportion of HBV patients newly diagnosed with ML increased (Table 2 and Appendix A).

Mature B cell lymphoma increased 3.8 times, from 389 in 2003 to 1492 in 2016. Mature T cell and NK-cell lymphoma increased about 6.2 times, from 26 to 161 in that period. Hodgkin lymphoma increased 1.9 times, from 24 in 2003 to 46 in 2016 (Table 2, Figure 1). The age of CHB patients who were first diagnosed with ML increased during the study period. The median age of CHB patients first diagnosed with ML was 52 in 2003, 53 in 2012, and 55 in 2016 (Appendix A).

#### 3.1.2. Association between Malignant Lymphoma Development and Specific Antiviral Agents

From 2012 to 2014, the incidence of newly diagnosed ML in treatment-naïve CHB patients initially treated with TDF or ETV increased. The incidence trend of ML decreased in the TDF-treated group compared with the ETV-treated group in the 3-year follow-up period. However, there was no statistically significant reduction in the TDF group (HR 0.84, CI 0.50–1.42, *p* = 0.527). CHB patients without co-morbidity showed a slightly decreased pattern in the incidence of ML in the TDF-treated group but the reduction was not statistically significant (HR 0.49, CI 0.20–1.21, *p* = 0.121) (Appendix A).

### 3.2. Chronic Hepatitis B as a Prognostic Factor of Malignant Lymphoma

#### 3.2.1. Survival Rate of Malignant Lymphoma with Chronic Hepatitis B

Chronic hepatitis B patients: A total of 13,942 CHB patients was diagnosed with newly developed ML from 2003 to 2016. Every CHB patient was observed for 5 years after the date of first diagnosis of ML. The 2-year survival rate of all patients was 76.8%, and the 5-year survival rate was 69.8% (Figure 2A). Mature B cell lymphoma with HBV had a 2-year survival rate of 76.9% and a 5-year survival rate of 70.1%. Mature T cell and NK-cell lymphoma had a 2-year of 72.0% and a 5-year of 63.7%. Hodgkin lymphoma had a 2-year of 84.6% and a 5-year of 77.4% (Figure 2B).

Cirrhotic patients with HBV. The survival rate of CHB ML patients with LC at baseline was significantly lower than in those without LC and even lower in decompensation. In patients without LC, the survival rates were 1 yr—84.1%, 3 yr—73.4%, and 5 yr—68.1%. The survival rates of CHB ML patients with LC were 1 yr—77.4%, 3 yr—64.9%, and 5 yr—59.2%. In decompensated LC, the survival rates were 1 yr—63.1%, 3 yr—48.2%, and 5 yr—40.0% (Figure 3A). These results were similar in subgroup analyses. In mature B cell lymphoma with LC, the survival rate was lower than without LC (Figure 3B). In mature T cell and NK cell lymphoma, the survival rates also were lower in advanced liver disease (Figure 3C).

#### 3.2.2. Prognosis of Malignant Lymphoma According to Chronic Hepatitis B Viral Activity

Chronic hepatitis B patients. The survival rate of CHB patients taking antivirals due to high viral activity prior to a diagnosis of ML was significantly lower than who did not take antivirals. CHB patients who started antivirals within the first 3 months before the diagnosis of ML were excluded because of the possibility of prophylactic antiviral therapy. The survival rates of ML patients taking antivirals were 1 yr—77.3%, 3 yr—64.5%, and 5 yr—58.3%. However, the survival rates who did not take antivirals were 1 yr—84.0%, 3 yr—73.4%, and 5 yr—68.0%. This result did not change even when controlling for age, sex, DM and HTN. The survival reduction in the antiviral group was statistically significant (HR 1.28, CI, 1.13–1.45, *p* < 0.001) (Appendix A, Figure 4A).

Cirrhotic patients with HBV. Cirrhotic patients who took antivirals before a diagnosis of ML had a worse prognosis than those who did not. The survival rates of LC with high viral activity (HBV DNA elevation) enough to take antivirals before ML diagnosis were 1 yr—75.5%, 3 yr—62.7%, and 5 yr—56.8%. However, the survival rates of LC who did not take antivirals at baseline (started antiviral therapy after a diagnosis of ML) were 1 yr—87.1%, 3 yr—68.5%, and 5 yr—62.1%. The reduced survival rate in the antiviral treatment group of LC was statistically significant (HR 0.67, CI, 0.48–0.94, *p* < 0.022) (Appendix A, Figure 4B).

## 4. Discussion

In the results of this NHIS data analysis, newly diagnosed ML in HBV patients increased continually. This phenomenon might be associated with the use of potent antiviral drugs and prolonged survival. However, the survival rate was lower who took antiviral agents to treat high viral activity before ML diagnosis than who did not need antivirals. In other words, viral activity high enough to need antiviral therapy could be a poor prognostic factor. Therefore, physicians need to be aware of the risk of ML in CHB patients and the importance of viral activity in determining the prognosis of HBV patients diagnosed with ML.

In the 1980s, Korea was classified as high prevalence area for HBV because more than 8% of people were infected with HBV. With the successful implementation of the national HBV vaccination program, the positivity for HBs Ag decreased steadily among young people [9]. However, HBV prevalence is still high enough (more than 3.0%) for Korea classified as an intermediate area. Therefore, almost all patients admitted to the hospital receive a screening test for HBV. After introduction of potent antivirals, the survival rates of CHB patients have increased continually [3], which increased the total number of CHB patients, as shown in our data. As the life expectancy of CHB patients has increased, various co-morbidities and extrahepatic malignancy became an important concern [4]. Asian data demonstrated that the incidence of extrahepatic malignancy in CHB patients was higher than general population [5,10]. Especially in cases of ML, genetic evidence revealed HBV as a risk factor [11,12]. The main evidence was the integration of HBV DNA in lymphoid tissue. As a result, the role of HBV integration and oncogenesis had been considered as a promising treatment mechanism [13]. It is well known that the risk of HBV integration continues to exist in CHB patients taking antiviral drugs [14]. Therefore, HBV DNA integration could occur in lymphoid tissue even taking antivirals.

In this study, we confirmed that both the occurrence of ML affected by HBV infection and its prognosis affected by CHB viral activity [15]. HBs Ag-positive ML were already known to have a poor prognosis [10]. In addition, HBs Ag negative and anti-HBc positive ML had a lower survival rate than anti-HBc negative [16]. In other words, patients exposed to HBV and suspected to have a covalently closed circular DNA, which carries a high probability of HBV DNA integration, had a worse prognosis. Our study revealed that ML patients taking antivirals due to high CHB viral activity, suggesting a possibility of the higher risk of HBV DNA integration, had lower survival rates.

Cirrhosis was a poor prognostic factor for ML with HBV. In decompensated LC, the survival rate was worse. Therefore, HBV-related LC and hepatic impairment are clinically important prognostic factors in ML. Similarly, viral activity (taking antiviral drugs) was also an independent prognostic factor of poor outcomes in LC patients. Probably, HBV DNA integration and genetic alteration affect the aggressiveness as well as the occurrence of ML in CHB patients [17].

A recent big data reported that TDF reduced HCC incidence better than ETV [18]. However, another multicenter study reported no difference [19]. In our study, the incidence of newly diagnosed ML did not differ between TDF and ETV. Therefore, physicians might not need to worry about reducing the risk of ML when selecting an antiviral drug. Because this study was conducted using records from the NHIS, there is a limitation of analysis. In addition, some patients could have been taking antiviral agents without insurance claim. Nonetheless, this study offers novel data about the natural course of all CHB patients with ML.

In conclusion, the viral activity could independently influence the prognosis of ML. In addition, cirrhotic patients had a poor survival rate after a diagnosis of ML, and even worse in decompensation. Clinically, our data suggest that the kinds of antivirals did not affect the risk of ML.

## Figures and Tables

**Figure 1 viruses-14-01943-f001:**
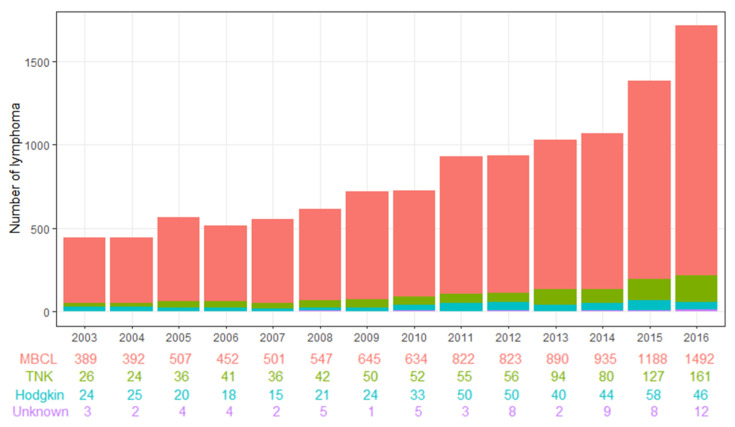
Annual incidence of malignant lymphoma subtypes in patients with hepatitis B virus. Note: MBCL, Mature B cell lymphoma; TNK, Mature T cell and NK-cell lymphoma; Hodgkin, Hodgkin lymphoma; Unknown, Unknown type of lymphoid neoplasm.

**Figure 2 viruses-14-01943-f002:**
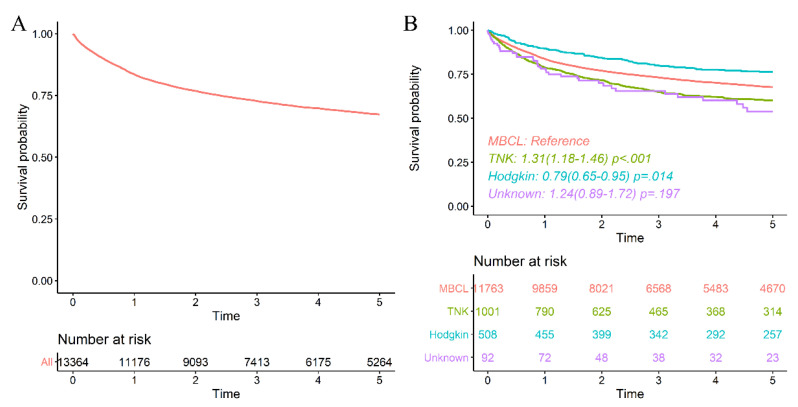
Survival rate of malignant lymphoma patients with chronic hepatitis B virus. (**A**). Overall survival of malignant lymphoma with chronic hepatitis B virus. (**B**). Survival rate of malignant lymphoma with chronic hepatitis B classified by subtype.

**Figure 3 viruses-14-01943-f003:**
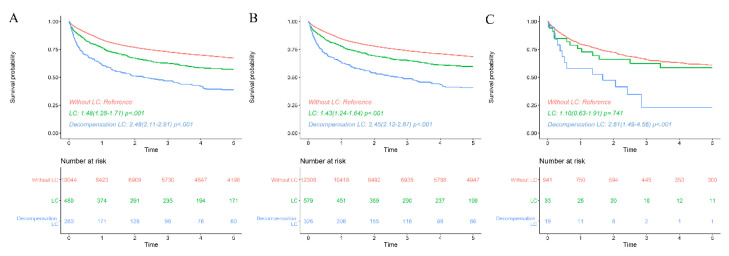
Survival rate of malignant lymphoma patients with chronic hepatitis B virus-diagnosed liver cirrhosis with compensation or decompensation. (**A**). The survival rate of patients with liver cirrhosis at their initial diagnosis of malignant lymphoma was significantly lower than that of patients without cirrhosis, and it was even lower in patients with decompensated liver cirrhosis. (**B**) (Mature B cell lymphoma), (**C**) (Mature T cell and NK-cell lymphoma). Subgroup analyses for each subtype showed a similar survival pattern.

**Figure 4 viruses-14-01943-f004:**
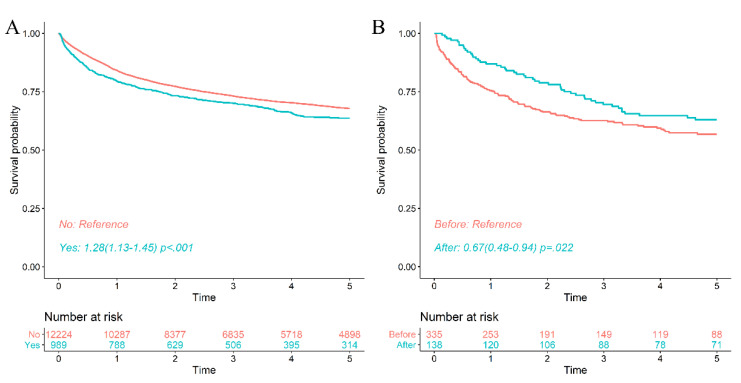
Poor prognosis when taking antivirals for chronic hepatitis B virus before a diagnosis of malignant lymphoma. (**A**). The survival rate of patients taking antiviral drugs before a diagnosis of malignant lymphoma was significantly lower than that of patients not taking antiviral drugs. (Patients who started antiviral drugs within the first 3 months before their diagnosis of malignant lymphoma were excluded). (**B**). The survival rate of cirrhotic patients taking antiviral drugs before their diagnosis of malignant lymphoma was significantly lower than that of patients not taking antiviral drugs. (After the diagnosis of malignant lymphoma, antiviral treatment was started. The patients who started antiviral drugs within 3 months before their diagnosis of malignant lymphoma were excluded).

**Table 1 viruses-14-01943-t001:** Characteristics of HBV patients with and without malignant lymphoma (2003–2016).

Variable	Group	No Lymphoma (n = 13,942)		Lymphoma (n = 13,942)		OR	95%CI		*p*
		n	%	n	%				
Sex	Male	8173	58.6%	8173	58.6%				
	Female	5769	41.4%	5769	41.4%	1.000	0.953	1.049	1.000
Age	<40	2500	17.9%	2500	17.9%				
	40–49	2783	20.0%	2783	20.0%	1.000	0.926	1.079	1.000
	50–59	3513	25.2%	3513	25.2%				
	>60	5146	36.9%	5146	36.9%				
Disease	Cirrhosis	1025	7.4%	939	6.7%	0.910	0.830	0.998	0.044
	Diabetes mellitus	5937	42.6%	6070	43.5%	1.040	0.992	1.090	0.108
	Hypertension	4097	29.4%	4119	29.5%	1.008	0.957	1.061	0.773
	CKD	396	2.8%	434	3.1%	1.099	0.957	1.262	0.181
Drugs	Spironolactone	477	3.4%	421	3.0%	0.879	0.769	1.004	0.058
	Terlipressin	177	1.3%	76	0.5%	0.426	0.324	0.556	0.000
	Somatostatin	70	0.5%	45	0.3%	0.642	0.438	0.930	0.020
	Propranolol	1821	13.1%	1731	12.4%	0.944	0.879	1.012	0.106
Antiviral agents	No	12,747	91.4%	12,793	91.8%				
	Yes	1195	8.6%	1149	8.2%	0.958	0.880	1.043	0.321

CKD, Chronic kidney disease.

**Table 2 viruses-14-01943-t002:** Crude incidence of malignant lymphoma in HBV patients in Korea (2003–2016).

Year	Malignant Lymphoma				Total
	MBCL	TNK	Hodgkin	Unknown	
2003	389	26	24	3	442
2004	392	24	25	2	443
2005	507	36	20	4	567
2006	452	41	18	4	515
2007	501	36	15	2	554
2008	547	42	21	5	615
2009	645	50	24	1	720
2010	634	52	33	5	724
2011	822	55	50	3	930
2012	823	56	50	8	937
2013	890	94	40	2	1026
2014	935	80	44	9	1068
2015	1188	127	58	8	1381
2016	1492	161	46	12	1711

MBCL, Mature B cell lymphoma; TNK, Mature T cell and NK-cell lymphoma; Hodgkin, Hodgkin lymphoma; Unknown, Unknown type of lymphoid neoplasm.

## Data Availability

Data sharing not applicable to this article, as no data are publicly available due to privacy or ethical restrictions.

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
