# Peer review of "Chronic Hepatitis B Viral Activity Enough to Take Antiviral Drug Could Predict the Survival Rate in Malignant Lymphoma"

_viruses, 2022, doi:10.3390/v14091943_

Round 1

Reviewer 1 Report

This is a very interesting epidemiologic study with some limitations that was underlined by the authors.

The main issue not addressed was vaccination for HBV in healthy population. Does HBV vaccination active in authors' country? In Europe HBV vaccination has drammatically reduced the incidence of infection. 

Author Response

Thank you very much for giving us an opportunity for revision.

Reviewer #1:

Comments to the author:

This is a very interesting epidemiologic study with some limitations that was underlined by the authors.

The main issue not addressed was vaccination for HBV in healthy population. Does HBV vaccination active in authors' country? In Europe HBV vaccination has dramatically reduced the incidence of infection.

Answer: Thank you for your kind comments. We added the comments of “National HBV Vaccination Program” in discussion.

  • With the successful implementation of the national HBV vaccination program, the positivity for HBs Ag decreased steadily among young people. (line 220)

Reviewer 2 Report

The paper by Kwang Il Seo et al is reporting interesting new information on the association between CHB and ML. One of their main finding regards the survival rate of patients taking antivirals due to high viral activity before their diagnosis with ML, that was significantly lower than that of patients with lower viral activity without antivirals (1 yr-77.3%, 3 yr-64.5%, and 5 yr-58.3% vs. 1 yr-84.0%, 3 yr-73.4%, and 5 24 yr-68.0%, respectively).

According to Figure 4 (A) 989 of the 13,213 CHB patients analyzed in this study were already taking antivirals at the time of ML diagnosis. The fact that only 7.5% of HBV infected patients were treated represents a major issue to correlate antiviral treatment status and overall survival.  According to the Authors, the reason for the lack of antiviral treatment is attributed to their “viral activity and related liver status” that, according to local guidelines, did not allow for antiviral treatment prescription (HBeAg-negative CHB patients with HBV DNA ≥2,000 IU/mL if serum ALT level is ≥2 times the ULN in HBeAg negative individuals). Since it is unlikely that 92.5% of the patients with HBV chronic infection diagnosed with ML were without CHB, the smaller proportion of treated patients is likely representing patients with more active and advanced liver disease. A more active disease represents by itself a condition associated with an higher mortality risk. Accordingly, the survival rate of CHB ML patients with cirrhosis at baseline was significantly lower than in those without cirrhosis, among whom 335/473 (60.8%) patients were already on therapy at the time of ML diagnosis. Altogether these data suggest that it is the liver disease stage the major determinant of the survival rates, and this does not come out clear from the paper.

To better investigate the reason for differences in mortality rates,  the survival analysis should be performed also by liver related and non liver related causes (ML and other..).

In the Discussion (lines 239-240) the sentence: “Our study revealed that ML patients taking antivirals due to high CHB viral activity, suggesting a high risk of HBV DNA integration, had lower survival rates” sounds cryptic (probably there is something missing). In any case, it is not clear yet whether antiviral therapy with NUCs, can prevent or not the integration of HBV DNA sequences into hepatocytes and/or lymphoid tissue. Therefore it is highly speculative to hypothesize that NUC therapy was not able to lower the risk of developing ML due to integration in lymphoid tissues.

Author Response

Thank you very much for giving us an opportunity for revision.

Reviewer #2:

Comments to the author:

The paper by Kwang Il Seo et al is reporting interesting new information on the association between CHB and ML. One of their main finding regards the survival rate of patients taking antivirals due to high viral activity before their diagnosis with ML, that was significantly lower than that of patients with lower viral activity without antivirals (1 yr-77.3%, 3 yr-64.5%, and 5 yr-58.3% vs. 1 yr-84.0%, 3 yr-73.4%, and 5 24 yr-68.0%, respectively).

  1. According to Figure 4 (A) 989 of the 13,213 CHB patients analyzed in this study were already taking antivirals at the time of ML diagnosis. The fact that only 7.5% of HBV infected patients were treated represents a major issue to correlate antiviral treatment status and overall survival. According to the Authors, the reason for the lack of antiviral treatment is attributed to their “viral activity and related liver status” that, according to local guidelines, did not allow for antiviral treatment prescription (HBeAg-negative CHB patients with HBV DNA ≥2,000 IU/mL if serum ALT level is ≥2 times the ULN in HBeAg negative individuals).

Answer: Thank you for your kind comments.

As you know, HBV is the most important cause of hepatocellular carcinoma (HCC) in the world. However, according to the 2021 hepatocellular carcinoma fact sheet in Korea, it is known that only 20% of HBV patients treated with antiviral therapy prior to diagnosis of HCC in 2008 and 31% in 2018. Therefore, the fact that 7.5% of patients were treated with HBV antiviral before the diagnosis of ML is clinically acceptable. Rather, considering the strong causal relationship between HBV and HCC, the fact that 7.5% of HBV patients took antiviral drugs before the diagnosis of ML is not a low ratio.

  1. Since it is unlikely that 92.5% of the patients with HBV chronic infection diagnosed with ML were without CHB, the smaller proportion of treated patients is likely representing patients with more active and advanced liver disease. A more active disease represents by itself a condition associated with a higher mortality risk. Accordingly, the survival rate of CHB ML patients with cirrhosis at baseline was significantly lower than in those without cirrhosis, among whom 335/473 (60.8%) patients were already on therapy at the time of ML diagnosis. Altogether these data suggest that it is the liver disease stage the major determinant of the survival rates, and this does not come out clear from the paper.

To better investigate the reason for differences in mortality rates, the survival analysis should be performed also by liver related and non-liver related causes (ML and other..).

Answer: Thank you for your precise comments. We absolutely agree with your opinion that the liver disease stage is the major determinant of the survival rates.

Therefore, we checked whether liver cirrhosis was present or not in order to classify liver disease status. The survival rate according to the presence or absence of liver cirrhosis is shown in figure 4B. As you see, the survival rate of cirrhotic patients taking antiviral drugs before their diagnosis of malignant lymphoma was significantly lower than that of patients not taking antiviral drugs. In other words, chronic hepatitis B viral activity enough to take antiviral drug could predict the survival rate in liver cirrhosis with Malignant Lymphoma.

  1. In the Discussion (lines 239-240) the sentence: “Our study revealed that ML patients taking antivirals due to high CHB viral activity, suggesting a high risk of HBV DNA integration, had lower survival rates” sounds cryptic (probably there is something missing). In any case, it is not clear yet whether antiviral therapy with NUCs, can prevent or not the integration of HBV DNA sequences into hepatocytes and/or lymphoid tissue. Therefore, it is highly speculative to hypothesize that NUC therapy was not able to lower the risk of developing ML due to integration in lymphoid tissues.

Answer: Thank you for your accurate comments. We totally agree with your opinion that we expressed our hypotheses too definitively. It was a pure effort to understand pathophysiology from the point of view of a clinician. We changed the expression as below.

  • Our study revealed that ML patients taking antivirals due to high CHB viral activity, suggesting a possibility of the higher risk of HBV DNA integration, had lower survival rates. (line 241)

Round 2

Reviewer 2 Report

 I can say that they have well explained the reason for the low percentage of HBV patients treated with antivirals, which was my main concern. Even though the risk of biased interpretations remains, there is no chance for them to get an accurate definition of the liver disease stage in untreated HBV patients. Nevertheless, I feel that there is enough value in the new information provided overall that deserves publication.